# Multi-ancestry study of blood lipid levels identifies four loci interacting with physical activity

Tuomas O. Kilpeläinen et al.[#]

Many genetic loci affect circulating lipid levels, but it remains unknown whether lifestyle factors, such as physical activity, modify these genetic effects. To identify lipid loci interacting with physical activity, we performed genome-wide analyses of circulating HDL cholesterol, LDL cholesterol, and triglyceride levels in up to 120,979 individuals of European, African, Asian, Hispanic, and Brazilian ancestry, with follow-up of suggestive associations in an additional 131,012 individuals. We find four loci, in/near CLASP1, LHX1, SNTA1, and CNTNAP2, that are associated with circulating lipid levels through interaction with physical activity; higher levels of physical activity enhance the HDL cholesterol-increasing effects of the CLASP1, LHX1, and SNTA1 loci and attenuate the LDL cholesterol-increasing effect of the CNTNAP2 locus. The CLASP1, LHX1, and SNTA1 regions harbor genes linked to muscle function and lipid metabolism. Our results elucidate the role of physical activity interactions in the genetic contribution to blood lipid levels.

Correspondence and requests for materials should be addressed to T.O.K. (email: tuomas.kilpelainen@sund.ku.dk) or to D.C.R. (email: rao@wustl.edu) or to R.J.F.L. (email: ruth.loos@mssm.edu). [#]A full list of authors and their affiliations appears at the end of the paper.

C irculating levels of blood lipids are strongly linked to the risk of atherosclerotic cardiovascular disease. Regular physical activity (PA) improves blood lipid profile by increasing the levels of high-density lipoprotein cholesterol (HDL-C) and decreasing the levels of low-density lipoprotein cholesterol (LDL-C) and triglycerides (TG)[1]. However, there is individual variation in the response of blood lipids to PA, and twin studies suggest that some of this variation may be due to genetic differences[2]. The genes responsible for this variability remain unknown.

More than 500 genetic loci have been found to be associated with blood levels of HDL-C, LDL-C, or TG in published genome-wide association studies (GWAS)[3–12]. At present, it is not known whether any of these main effect associations are modified by PA. Understanding whether the impact of lipid loci can be modified by PA is important because it may give additional insight into biological mechanisms and identify subpopulations in whom PA is particularly beneficial.

Here, we report results from a genome-wide meta-analysis of gene–PA interactions on blood lipid levels in up to 120,979 adults of European, African, Asian, Hispanic, or Brazilian ancestry, with follow-up of suggestive associations in an additional 131,012 individuals. We show that four loci, in/near CLASP1, LHX1, SNTA1, and CNTNAP2, are associated with circulating lipid levels through interaction with PA. None of these four loci have been identified in published main effect GWAS of lipid levels. The CLASP1, LHX1, and SNTA1 regions harbor genes linked to muscle function and lipid metabolism. Our results elucidate the role of PA interactions in the genetic contribution to blood lipid levels.

## Results

**Genome-wide interaction analyses in up to 250,564 individuals.** We assessed effects of gene–PA interactions on serum HDL-C, LDL-C, and TG levels in 86 cohorts participating in the Cohorts for Heart and Aging Research in Genomic Epidemiology (CHARGE) Gene-Lifestyle Interactions Working Group[13]. PA was harmonized across participating studies by categorizing it into a dichotomous variable. The participants were defined as inactive if their reported weekly energy expenditure in moderate-to-vigorous intensity leisure-time or commuting PA was less than 225 metabolic equivalent (MET) minutes per week (corresponding to approximately 1 h of moderate-intensity PA), while all other participants were defined as physically active (Supplementary Data 1).

The analyses were performed in two stages. Stage 1 consisted of genome-wide meta-analyses of linear regression results from 42 cohorts, including 120,979 individuals of European [$n = 84,902$], African [$n = 20,487$], Asian [$n = 6403$], Hispanic [$n = 4749$], or Brazilian [$n = 4438$] ancestry (Supplementary Tables 1 and 2; Supplementary Data 2; Supplementary Note 1). All variants that reached two-sided $P < 1 \times 10^{-6}$ in the Stage 1 multi-ancestry meta-analyses or ancestry-specific meta-analyses were taken forward to linear regression analyses in Stage 2, which included 44 cohorts and 131,012 individuals of European [$n = 107,617$], African [$n = 5384$], Asian [$n = 6590$], or Hispanic [$n = 11,421$] ancestry (Supplementary Tables 3 and 4; Supplementary Data 3; Supplementary Note 2). The summary statistics from Stage 1 and Stage 2 were subsequently meta-analyzed to identify lipid loci whose effects are modified by PA.

We identified lipid loci interacting with PA by three different approaches applied to the meta-analysis of Stage 1 and Stage 2: (i) we screened for genome-wide significant SNP × PA-interaction effects ($P_{\text{INT}} < 5 \times 10^{-8}$); (ii) we screened for genome-wide significant 2 degree of freedom (2df) joint test of SNP main

effect and SNP × PA interaction[14] ($P_{\text{JOINT}} < 5 \times 10^{-8}$); and (iii) we screened all previously known lipid loci[3–12] for significant SNP × PA-interaction effects, Bonferroni-correcting for the number of independent variants tested ($r^2 < 0.1$ within 1 Mb distance; $P_{\text{INT}} = 0.05/501 = 1.0 \times 10^{-4}$).

**PA modifies the effect of four loci on lipid levels.** Three novel loci (>1 Mb distance and $r^2 < 0.1$ with any previously identified lipid locus) were identified: in CLASP1 (rs2862183, $P_{\text{INT}} = 8 \times 10^{-9}$), near LHX1 (rs295849, $P_{\text{INT}} = 3 \times 10^{-8}$), and in SNTA1 (rs141588480, $P_{\text{INT}} = 2 \times 10^{-8}$), which showed a genome-wide significant SNP × PA interaction on HDL-C in all ancestries combined (Table 1, Figs. 1–4). Higher levels of PA enhanced the HDL cholesterol-increasing effects of the CLASP1, LHX1, and SNTA1 loci. A novel locus in CNTNAP2 (rs190748049) was genome-wide significant in the joint test of SNP main effect and SNP × PA interaction ($P_{\text{JOINT}} = 4 \times 10^{-8}$) and showed moderate evidence of SNP × PA interaction ($P_{\text{INT}} = 2 \times 10^{-6}$) in the meta-analysis of LDL-C in all ancestries combined (Table 1, Fig. 5). The LDL-C-increasing effect of the CNTNAP2 locus was attenuated in the physically active group as compared to the inactive group. None of these four loci have been identified in previous main effect GWAS of lipid levels.

**No interaction between known main effect lipid loci and PA.** Of the previously known 260 main effect loci for HDL-C, 202 for LDL-C, and 185 for TG[3–12], none reached the Bonferroni-corrected threshold (two-sided $P_{\text{INT}} = 1.0 \times 10^{-4}$) for SNP × PA interaction alone (Supplementary Data 4-6). We also found no significant interaction between a combined score of all published European-ancestry loci for HDL-C, LDL-C, or TG with PA (Supplementary Datas 7–9) using our European-ancestry summary results (two-sided $P_{\text{HDL-C}} = 0.14$, $P_{\text{LDL-C}} = 0.77$, and $P_{\text{TG}} = 0.86$, respectively), suggesting that the beneficial effect of PA on lipid levels may be independent of genetic risk[15].

**Potential functional roles of the loci interacting with PA.** While the mechanisms underlying the beneficial effect of PA on circulating lipid levels are not fully understood, it is thought that the changes in plasma lipid levels are primarily due to an improvement in the ability of skeletal muscle to utilize lipids for energy due to enhanced enzymatic activities in the muscle[16,17]. Of the four loci we found to interact with PA, three, in CLASP1, near LHX1, and in SNTA1, harbor genes that may play a role in muscle function[18,19] and lipid metabolism[20,21].

The lead variant rs2862183 (minor allele frequency (MAF) 22%) in the CLASP1 locus which interacts with PA on HDL-C levels is an intronic SNP in CLASP1 that encodes a microtubule-associated protein (Fig. 2). The rs2862183 SNP is associated with CLASP1 expression in esophagus muscularis ($P = 3 \times 10^{-5}$) and is in strong linkage disequilibrium ($r^2 > 0.79$) with rs13403769 variant that shows the strongest association with CLASP1 expression in the region ($P = 7 \times 10^{-7}$). Another potent causal candidate gene in this locus is the nearby GLI2 gene which has been found to play a role in skeletal myogenesis[18] and the conversion of glucose to lipids in mouse adipose tissue[20] by inhibiting hedgehog signaling.

The rs295849 (MAF 38%) variant near LHX1 interacts with PA on HDL-C levels. However, the more likely causal candidate gene in this locus is acetyl-CoA carboxylase (ACACA), which plays a crucial role in fatty acid metabolism[21] (Fig. 3). Rare acetyl-CoA carboxylase deficiency has been linked to hypotonic myopathy, severe brain damage, and poor growth[22].

The lead variant in the SNTA1 locus (rs141588480) interacts with PA on HDL-C and is an insertion only found in individuals

**Table 1 Lipid loci identified through interaction with physical activity ($P_{INT} < 5 \times 10^{-8}$) or through joint test for SNP main effect and SNP × physical activity interaction ($P_{JOINT} < 5 \times 10^{-8}$)**

| Trait | SNP | Chr:Pos | Gene | EA/OA | EAF | N inactive | N active | Beta$_{INT}$ | se$_{INT}$ | $P_{INT}$ | $P_{JOINT}$ |
|---|---|---|---|---|---|---|---|---|---|---|---|
| *Loci identified through interaction with physical activity* | | | | | | | | | | | |
| HDL-C | rs2862183 | 2:122415398 | *CLASP1* | T/C | 0.22 | 76,674 | 154,118 | 0.014 | 0.003 | 7.5E$^{-9}$ | 3.6E$^{-7}$ |
| HDL-C | rs295849 | 17:35161748 | *LHX1* | T/G | 0.38 | 78,288 | 160,924 | 0.009 | 0.002 | 2.7E$^{-8}$ | 6.8E$^{-7}$ |
| HDL-C | rs141588480 | 20:32013913 | *SNTA1* | Ins/Del | 0.95 | 8,694 | 18,585 | 0.054 | 0.010 | 2.0E$^{-8}$ | 6.1E$^{-7}$ |
| *Loci identified through joint test for SNP main effect and SNP × physical activity interaction* | | | | | | | | | | | |
| LDL-C | rs190748049 | 7:146418260 | *CNTNAP2* | C/T | 0.95 | 14,912 | 28,715 | −7.2 | 1.5 | 1.6E$^{-6}$ | 4.2E$^{-8}$ |

All loci were identified in the meta-analyses of all ancestries combined. HDL-C was natural logarithmically transformed, whereas LDL-C was not transformed. The *P* values are two-sided and were obtained using a meta-analysis of linear regression model results. *EA* effect allele, *EAF* effect allele frequency, *OA* other allele, *beta$_{INT}$* effect size for interaction with physical activity (=the change in logarithmically transformed HDL-C or untransformed LDL-C levels in the active group as compared to the inactive group per each effect allele), *se$_{INT}$* standard error for interaction with physical activity

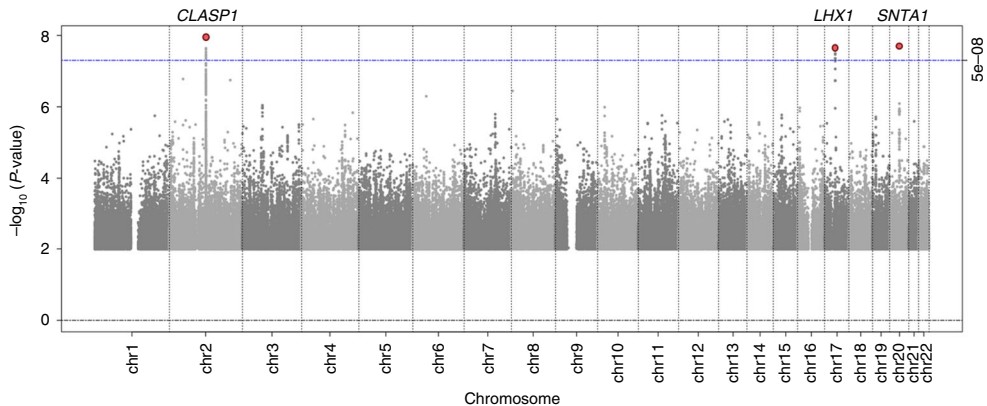

**Fig. 1** Genome-wide results for interaction with physical activity on HDL cholesterol levels. The *P* values are two-sided and were obtained by a meta-analysis of linear regression model results ($n$ up to 250,564). Three loci, in/near *CLASP1*, *LHX1*, and *SNTA1*, reached genome-wide significance ($P < 5 \times 10^{-8}$) as indicated in the plot

of African (MAF 6%) or Hispanic (MAF 1%) ancestry. The rs141588480 insertion is in the *SNTA1* gene that encodes the syntrophin alpha 1 protein, located at the neuromuscular junction and altering intracellular calcium ion levels in muscle tissue (Fig. 4). *Snta1*-null mice exhibit differences in muscle regeneration after a cardiotoxin injection[19]. Two weeks following the injection into mouse tibialis anterior, the muscle showed hypertrophy, decreased contractile force, and neuromuscular junction dysfunction. Furthermore, exercise endurance of the mice was impaired in the early phase of muscle regeneration[19]. In humans, *SNTA1* mutations have been linked to the long-QT syndrome[23].

The fourth locus interacting with PA is *CNTNAP2*, with the lead variant (rs190748049) intronic and no other genes nearby (Fig. 5). The rs190748049 variant is most common in African-ancestry (MAF 8%), less frequent in European-ancestry (MAF 2%), and absent in Asian- and Hispanic-ancestry populations. The protein coded by the *CNTNAP2* gene, contactin-associated protein like-2, is a member of the neurexin protein family. The protein is located at the juxtaparanodes of myelinated axons where it may have an important role in the differentiation of the axon into specific functional subdomains. Mice with a *Cntnap2* knockout are used as an animal model of autism and show altered phasic inhibition and a decreased number of interneurons[24]. Human *CNTNAP2* variants have been associated with risk of autism and related behavioral disorders[25].

**Joint test of SNP main effect and SNP × PA interaction.** We found 101 additional loci that reached genome-wide significance in the 2df joint test of SNP main effect and SNP × PA interaction

on HDL-C, LDL-C, or TG. However, none of these loci showed evidence of SNP × PA interaction ($P_{INT} > 0.001$) (Supplementary Data 10). All 101 main effect-driven loci have been identified in previous GWAS of lipid levels[3–12].

## Discussion

In this genome-wide study of up to 250,564 adults from diverse ancestries, we found evidence of interaction with PA for four loci, in/near *CLASP1*, *LHX1*, *SNTA1*, and *CNTNAP2*. Higher levels of PA enhanced the HDL cholesterol-increasing effects of *CLASP1*, *LHX1*, and *SNTA1* loci and attenuated the LDL cholesterol-increasing effect of the *CNTNAP2* locus. None of these four loci have been identified in previous main effect GWAS for lipid levels[3–12].

The loci in/near *CLASP1*, *LHX1*, and *SNTA1* harbor genes linked to muscle function[18,19] and lipid metabolism[20,21]. More specifically, the *GLI2* gene within the *CLASP1* locus has been found to play a role in myogenesis[18] as well as in the conversion of glucose to lipids in adipose tissue[20]; the *ACACA* gene within the *LHX1* locus plays a crucial role in fatty acid metabolism[21] and has been connected to hypotonic myopathy[22]; and the *SNTA1* gene is linked to muscle regeneration[19]. These functions may relate to differences in the ability of skeletal muscle to use lipids as an energy source, which may modify the beneficial impact of PA on blood lipid levels[16,17].

The inclusion of diverse ancestries in the present meta-analyses allowed us to identify two loci that would have been missed in meta-analyses of European-ancestry individuals alone. In particular, the lead variant (rs141588480) in the *SNTA1* locus is only polymorphic in African and Hispanic ancestries, and the lead

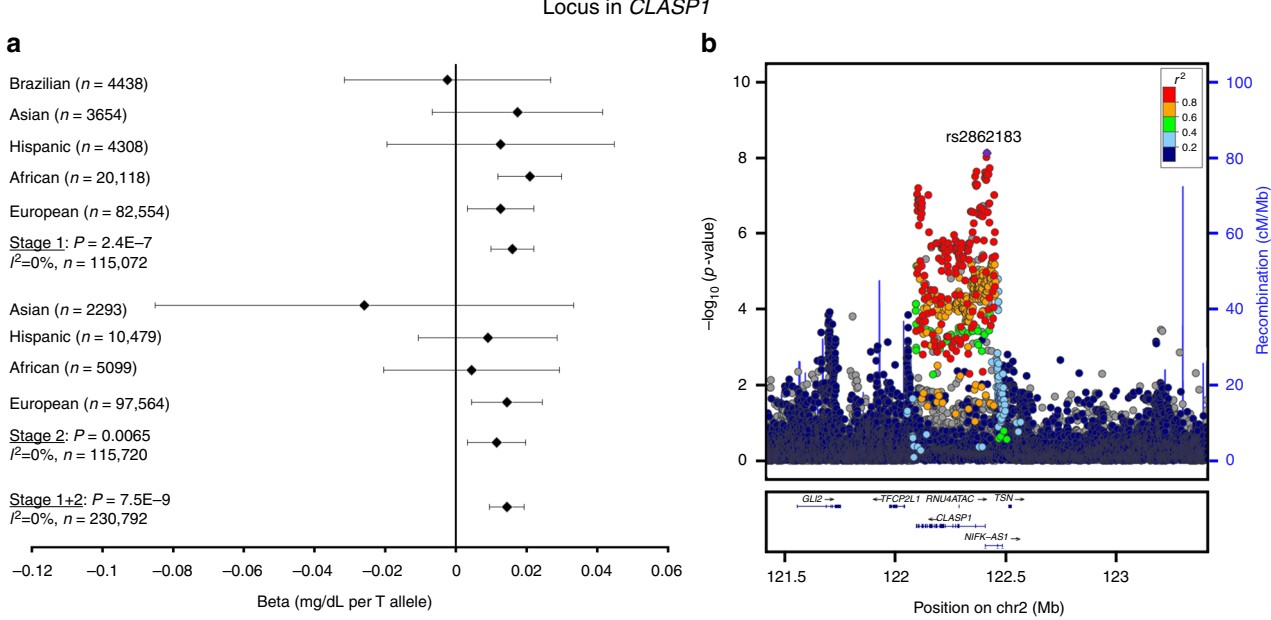

**Fig. 2** Interaction of rs2862183 in *CLASP1* with physical activity on HDL cholesterol levels. The beta and 95% confidence intervals in the forest plot (**a**) is shown for the rs2862183 × physical activity interaction term, i.e., it indicates the increase in logarithmically transformed HDL cholesterol levels in the active group as compared to the inactive group per each T allele of rs2862183. The $-\log_{10}(P$ value) in the association plot (**b**) is also shown for the rs2862183 × physical activity interaction term. The P values are two-sided and were obtained by a meta-analysis of linear regression model results. The figure was generated using LocusZoom (http://locuszoom.org)

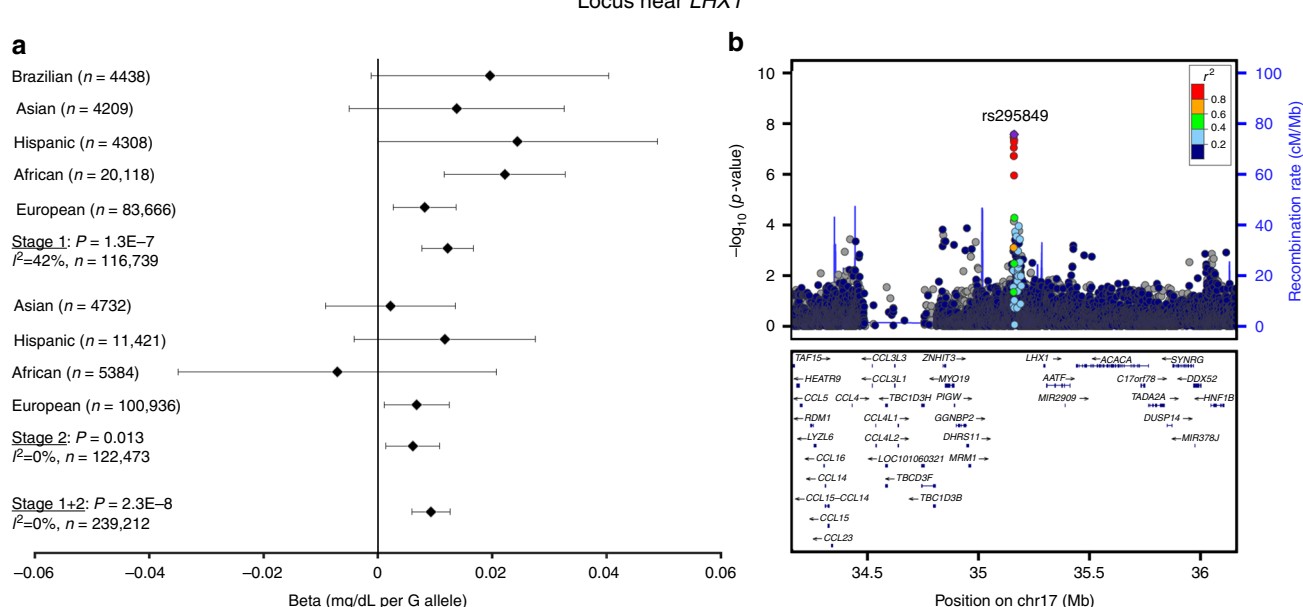

**Fig. 3** Interaction of rs295849 near *LHX1* with physical activity on HDL cholesterol levels. The beta and 95% confidence intervals in the forest plot (**a**) is shown for the rs295849 × physical activity interaction term, i.e., it indicates the increase in logarithmically transformed HDL cholesterol levels in the active group as compared to the inactive group per each G allele of rs295849. The $-\log_{10}$ (P value) in the association plot (**b**) is also shown for the rs295849 × physical activity interaction term. The P values are two-sided and were obtained by a meta-analysis of linear regression model results. The figure was generated using LocusZoom (http://locuszoom.org)

variant (rs190748049) in the *CNTNAP2* locus is four times more frequent in African-ancestry than in European-ancestry. Our findings highlight the importance of multi-ancestry investigations of gene-lifestyle interactions to identify novel loci.

We did not find additional novel loci when jointly testing for SNP main effect and interaction with PA. While 101 loci reached

genome-wide significance in the joint test on HDL-C, LDL-C, or TG, all of these loci have been identified in previous GWAS of lipid levels[3–12], and none of them showed evidence of SNP × PA interaction. The 2df joint test bolsters the power to detect novel loci when both main and an interaction effect are present[14]. The lack of novel loci identified by the 2df test suggests that the loci

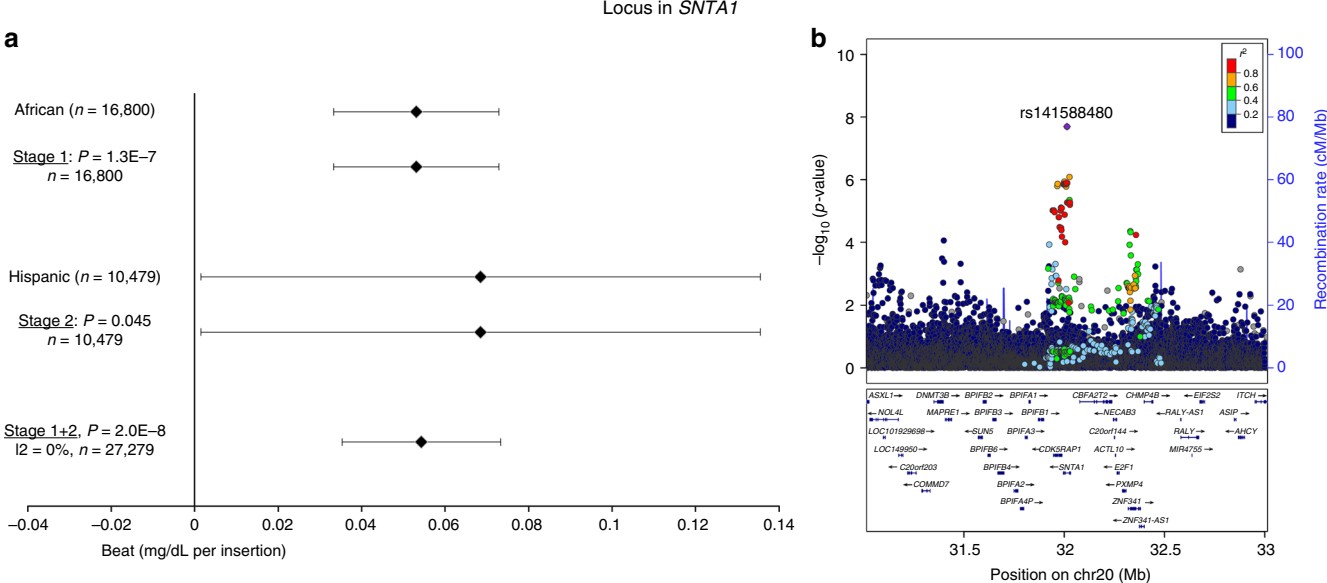

**Fig. 4** Interaction of rs141588480 in *SNTA1* with physical activity on HDL cholesterol levels. The beta and 95% confidence intervals in the forest plot (**a**) is shown for the rs141588480 × physical activity interaction term, i.e., it indicates the increase in logarithmically transformed HDL cholesterol levels in the active group as compared to the inactive group per each insertion of rs141588480. The −log₁₀ (*p* value) in the association plot (**b**) is also shown for the rs141588480 × physical activity interaction term. While the rs141588480 variant was identified in African-ancestry individuals in Stage 1, the variant did not pass QC filters in the Stage 2 African-ancestry cohorts, due to insufficient sample sizes of these cohorts. The *P* values are two-sided and were obtained by a meta-analysis of linear regression model results. The figure was generated using LocusZoom (http://locuszoom.org)

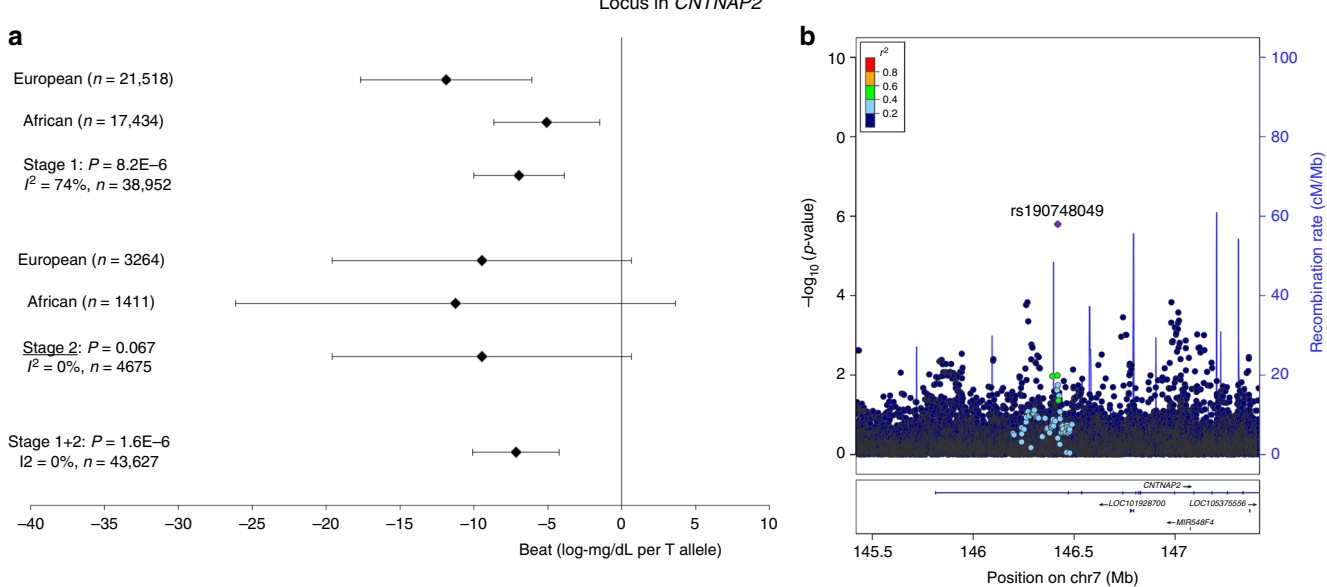

**Fig. 5** Interaction of rs190748049 variant in *CNTNAP2* with physical activity on LDL cholesterol levels. The rs190748049 variant was genome-wide significant in the joint test for SNP main effect and SNP × physical activity interaction and reached $P = 2 \times 10^{-6}$ for the SNP × physical activity interaction term alone. The beta and 95% confidence intervals in the forest plot (**a**) is shown for the SNP × physical activity interaction term, i.e., it indicates the decrease in LDL cholesterol levels in the active group as compared to the inactive group per each T allele of rs190748049. The −log₁₀ (*P* value) in the association plot (**b**) is also for the SNP × physical activity interaction term. The *P* values are two-sided and were obtained using a meta-analysis of linear regression model results. The figure was generated using LocusZoom (http://locuszoom.org)

showing the strongest SNP × PA interaction on lipid levels are not the same loci that show a strong main effect on lipid levels.

In summary, we identified four loci containing SNPs that enhance the beneficial effect of PA on lipid levels. The identification of the *SNTA1* and *CNTNAP2* loci interacting with PA was made possible by the inclusion of diverse ancestries in the analyses. The gene regions that harbor loci interacting with PA involve pathways targeting muscle function and lipid metabolism. Our findings elucidate the role and underlying mechanisms of PA interactions in the genetic regulation of blood lipid levels.

## Methods

**Study design**. The present study collected summary data from 86 participating cohorts and no individual-level data were exchanged. For each of the participating cohorts, the appropriate ethics review board approved the data collection and all participants provided informed consent.

We included men and women 18–80 years of age and of European, African, Asian, Hispanic, or Brazilian ancestry. The meta-analyses were performed in two stages[13]. Stage 1 meta-analyses included 42 studies with a total of 120,979 individuals of European ($n = 84,902$), African ($n = 20,487$), Asian ($n = 6403$), Hispanic ($n = 4749$), or Brazilian ancestry ($n = 4438$) (Supplementary Table 1; Supplementary Data 2; Supplementary Note 1). Stage 2 meta-analyses included 44 studies with a total of 131,012 individuals of European ($n = 107,617$), African ($n = 5384$), Asian ($n = 6590$), or Hispanic ($n = 11,421$) ancestry (Supplementary Table 3; Supplementary Data 3; Supplementary Note 2). Studies participating in Stage 1 meta-analyses carried out genome-wide analyses, whereas studies participating in Stage 2 only performed analyses for 17,711 variants that reached $P < 10^{-6}$ in the Stage 1 meta-analyses and were observed in at least two different Stage 1 studies with a pooled sample size > 4000. The Stage 1 and Stage 2 meta-analyses were performed in all ancestries combined and in each ancestry separately.

**Outcome traits: LDL-C, HDL-C, and TG**. The levels of LDL-C were either directly assayed or derived using the Friedewald equation (if TG ≤ 400 mg dl$^{-1}$ and fasting ≥ 8 h). We adjusted LDL-C levels for lipid-lowering drug use if statin use was reported or if unspecified lipid-lowering drug use was listed after 1994, when statin use became common. For directly assayed LDL-C, we divided the LDL-C value by 0.7. If LDL-C was derived using the Friedewald equation, we first adjusted total cholesterol for statin use (total cholesterol divided by 0.8) before the usual calculation. If study samples were from individuals who were nonfasting, we did not include either TG or calculated LDL-C in the present analyses. The HDL-C and TG variables were natural log-transformed, while LDL-C was not transformed.

**PA variable**. The participating studies used a variety of ways to assess and quantify PA (Supplementary Data 1). To harmonize the PA variable across all participating studies, we coded a dichotomous variable, inactive vs. active, that could be applied in a relatively uniform way in all studies, and that would be congruent with previous findings on SNP × PA interactions[26–28] and the relationship between PA and disease outcomes[29]. Inactive individuals were defined as those with <225 MET-min per week of moderate-to-vigorous leisure-time or commuting PA ($n = 84,495$; 34% of all participants) (Supplementary Data 1). We considered all other participants as physically active. In studies where MET-min per week measures of PA were not available, we defined inactive individuals as those engaging in ≤1 h/week of moderate-intensity leisure-time PA or commuting PA. In studies with PA measures that were not comparable to either MET-min or hours/week of PA, we defined the inactive group using a percentage cut-off, where individuals in the lowest 25% of PA levels were defined as inactive and all other individuals as active.

**Genotyping and imputation**. Genotyping was performed by each participating study using Illumina or Affymetrix arrays. Imputation was conducted on the cosmopolitan reference panel from the 1000 Genomes Project Phase I Integrated Release Version 3 Haplotypes (2010–2011 data freeze, 2012-03-14 haplotypes). Only autosomal variants were considered. Specific details of each participating study's genotyping platform and imputation software are described in Supplementary Tables 2 and 4.

**Quality control**. The participating studies excluded variants with MAF < 1%. We performed QC for all study-specific results using the EasyQC package in $R$[30]. For each study-specific results file, we filtered out genetic variants for which the product of minor allele count (MAC) in the inactive and active strata and imputation quality [min(MAC$_{\text{INACTIVE}}$,MAC$_{\text{ACTIVE}}$) × imputation quality] did not reach 20. This removed unstable study-specific results that reflected small sample size, low MAC, or low-imputation quality. In addition, we excluded all variants with imputation quality measure <0.5. To identify issues with relatedness, we examined QQ plots and genomic control inflation lambdas in each study-specific results file as well as in the meta-analysis results files. To identify issues with allele frequencies, we compared the allele frequencies in each study file against ancestry-specific allele frequencies in the 1000 Genomes reference panel. To identify issues with trait transformation, we plotted the median standard error against the maximal sample size in each study. The summary statistics for all beta-coefficients, standard errors, and $P$ values were visually compared to observe discrepancies. Any issues that were found during the QC were resolved by contacting the analysts from the participating studies. Additional details about QC in the context of interactions, including examples, may be found elsewhere[13].

**Analysis methods**. All participating studies used the following model to test for interaction:

$$E[Y] = \beta_0 + \beta_E * PA + \beta_G * G + \beta_{\text{INT}} * G * PA + \boldsymbol{\beta_c} * \boldsymbol{C},$$

where $Y$ is the HDL-C, LDL-C, or TG value, $PA$ is the PA variable with 0 or 1 coding for active or inactive group, and $G$ is the dosage of the imputed genetic variant coded additively from 0 to 2. The $C$ is the vector of covariates which included age, sex, study center (for multi-center studies), and genome-wide principal components. From this model, the studies provided the estimated genetic main effect ($\beta_G$), estimated interaction effect ($\beta_{GE}$), and a robust estimate of the covariance between $\beta_G$ and $\beta_{GE}$. Using these estimates, we performed inverse variance-weighted meta-analyses for the SNP × PA interaction term alone, and 2df joint meta-analyses of the SNP effect and SNP × PA interaction combined by the method of Manning et al.[14], using the METAL meta-analysis software. We applied genomic control correction twice in Stage 1, first for study-specific GWAS results and again for meta-analysis results, whereas genomic control correction was not applied to the Stage 2 results as interaction testing was only performed at select variants. We considered a variant that reached two-sided $P < 5 \times 10^{-8}$ in the meta-analysis for the interaction term alone or in the joint test of SNP main effect and SNP × PA interaction, either in the ancestry-specific analyses or in all ancestries combined, as genome-wide significant. The loci were defined as independent if the distance between the lead variants was >1 Mb.

**Combined PA-interaction effect of all known lipid loci**. To identify all published SNPs associated with HDL-C, LDL-C, or TG, we extended the previous curated list of lipid loci by Davis et al.[4] by searching PubMed and Google Scholar databases and screening the GWAS Catalog. After LD pruning by $r^2 < 0.1$ in the 1000 Genomes European-ancestry reference panel, 260 independent loci remained associated with HDL cholesterol, 202 with LDL cholesterol, and 185 with TG (Supplementary Datas 7–9). To approximate the combined PA interaction of all known European-ancestry loci associated with HDL-C, LDL-C, or TG, we calculated their combined interaction effect as the weighted sum of the individual SNP coefficients in our genome-wide summary results for European-ancestry. This approach has been described previously in detail by Dastani et al.[31] and incorporated in the package "gtx" in R. We did not weigh the loci by their main effect estimates from the discovery GWAS data.

**Examining the functional roles of loci interacting with PA**. We examined published associations of the identified lipid loci with other complex traits in genome-wide association studies by using the GWAS Catalog of the European Bioinformatics Institute and the National Human Genome Research Institute. We extracted all published genetic associations with $r^2 > 0.5$ and distance < 500 kb from the identified lipid-associated lead SNPs[32]. We also studied the cis-associations of the lead SNPs with all genes within ±1 Mb distance using the GTEx portal[33]. We excluded findings where our lead SNP was not in strong LD ($r^2 > 0.5$) with the peak SNP associated with the same gene transcript.

## Data availability

The meta-analysis summary results are available for download on the CHARGE dbGaP website under accession phs000930.

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

## Acknowledgments

The present work was largely supported by a grant from the US National Heart, Lung, and Blood Institute (NHLBI) of the National Institutes of Health (R01HL118305). The full list of acknowledgments appears in the Supplementary Notes 3 and 4.

## Author contributions

T.O.K., K. Schwander., D.C.R., and R.J.F.L. conceived and designed the study. The members of the writing group were T.O.K., A.R.B., R.N., Y.J.S., K.Schwander., T. Winkler, H.J., D.I.C., A. Manning., I.N., B.M.P., K.R., P.B.M., M.F., L.A.C., C.N.R., A. C.M., D.C.R., and R.J.F.L. The genome-wide association results were provided by A.R.B., R.N., Y.J.S., K.Strauch, T. Winkler, D.I.C., A. Manning., I.N., H.A., M.R.B., L.d.l.F., N.F., X.G., D.V., S.A., M.F.F., M.K., S.K.M., M. Richard, H.W., Z.W., T.M.B., L.F.B., A.C., R.D., V.F., F.P.H., A.R.V.R.H., C. Li, K.K.L., J.M., X.S., A.V.S., S.M.T., M. Alver, M. Amini, M. Boissel, J.F.C., X.C., J. Divers, E.E., C. Gao, M. Graff, S.E.H., M.H., F.C.H., A.U.J., J.H.Z., A.T.K., B.K., F.L., L.P.L., I.M.N., R. Rauramaa, M. Riaz, A.R., R. Rueedi, H.M.S., F.T., P.J. v.d.M., T.V.V., N.V., E.B.W., W.W., X.L., L.R.Y., N.A., D.K.A., E.B., M. Brumat, B.C., M.C., Y.D.I.C., M.P.C., J.C., R.d.M., H.J.d.S., P.S.d.V., A.D., J. Ding, C.B.E., J.D.F., Y.F., K.P.G., M. Ghanbari, F.G., C.C.G., D.G., T.B.H., J.H., S.H., C.K.H., S.C.H., A.I., J.B.J., W.P.K., P.K., J.E.K., S.B.K., Z.K., J.K., C.D.L., C. Langenberg, L.J.L., K.L., R.N.L., C.E.L., J. Liang, J. Liu, R.M., A. Manichaikul, T.M., A. Metspalu, Y.M., K.L.M., T.H.M., A.D.M., M.A.N., E.E.K.N., C.P.N., S.N., J.M.N., J.O., N.D.P., G.J.P., R.P., N.L.P., A. Peters, P.A.P., O.P., D.J.P., A. Poveda, O.T.R., S.S.R., N.R., J.G.R., L.M.R., I.R., P.J.S., R.A.S., S.S.S., M.S., J.A.S., H.S., T.S., J.M.S., B.S., K.St., H.T., K.D.T., M.Y.T., J.T., A.G.U., M.Y.v.d.E., D.v.H., T.V., M.W., P.W., G.W., Y.B.X., J.Y., C.Y., J.M.Y., W. Zhao, A.B.Z., D.M.B., M. Boehnke, D.W.B., U.d.F., I.J.D., P.E., T.E., B.I.F., P.F., P.G., C. Gieger, N.K., M.L., T.A.L., T.L., P.K.E.M., A.J.O., B.W.J.H.P., N.J.S., X.O.S., P.v.d.H., J.V.V.V.O., P.V., L.E.W., Y.X.W., N.J.W., D.R.W., T. Wu, W. Zheng, X.Z., M.K.E., P.W.F., V.G., C.H., B.L.H., T.N.K., Y.L., K.E.N., A.C.P., P.M.R., E.S.T., R.M.v.D., E.R.F., S.L.R.K., C.T.L., D.O.M.K., M.A.P., S.R., C.M.v.D., J.I.R., C.B.K., W.J.G., B.M.P., K.R., P.B.M., M.F., L.A.C., C.N.R., A.C.M., D.C.R., and R.J.F.L.; The meta-analyses were performed by T.O.K. and H.J.; The combined physical activity interaction effects of all known lipid loci were examined by T.O.K. and H.J.; T.O.K. and C.V.N. collected look-up information in GWAS studies for other traits; T.O.K. and C.V.N. carried out the eQTL look-ups. All authors reviewed and approved the final manuscript.

## Additional information

**Competing interests:** Bruce M. Psaty serves on the DSMB of a clinical trial funded by the manufacturer (Zoll LifeCor) and on the Steering Committee of the Yale Open Data Access Project funded by Johnson & Johnson. Brenda W.J.H. Penninx has received research funding (nonrelated to the work reported here) from Jansen Research and Boehringer Ingelheim. Mike A. Nalls' participation is supported by a consulting contract between Data Tecnica International and the National Institute on Aging, National Institutes of Health, Bethesda, MD, USA. Dr. Nalls also consults for Illumina Inc, the Michael J. Fox Foundation and University of California Healthcare among others, and has a Commercial affiliation with Data Technica International, Glen Echo, MD, USA. Jost B. Jonas serves as a consultant for Mundipharma Co. (Cambridge, UK), patent holder with Biocompatibles UK Ltd. (Franham, Surrey, UK) (Title: Treatment of eye diseases using encapsulated cells encoding and secreting neuroprotective factor and/or anti-angiogenic factor; Patent number: 20120263794), and is patent applicant with University of Heidelberg (Heidelberg, Germany) (Title: Agents for use in the therapeutic or prophylactic treatment of myopia or hyperopia; Europäische Patentanmeldung 15,000 771.4). Paul W. Franks has been a paid consultant in the design of a personalized Nutrition trial (PREDICT) as part of a private-public partnership at Kings College London, UK, and has received research support from several pharmaceutical Companies as part of European Union Innovative Medicines Initiative (IMI) Projects. Terho Lehtimäki is employed by Fimlab Ltd. Ozren Polasek is employed by Gen-info Ltd. The remaining authors declare no competing interests.

Tuomas O. Kilpeläinen[1,2], Amy R. Bentley[3], Raymond Noordam[4], Yun Ju Sung[5], Karen Schwander[5], Thomas W. Winkler[6], Hermina Jakupović[1], Daniel I. Chasman[7,8], Alisa Manning[9,10], Ioanna Ntalla[11], Hugues Aschard[12,13], Michael R. Brown[14], Lisa de las Fuentes[5,15], Nora Franceschini[16], Xiuqing Guo[17], Dina Vojinovic[18], Stella Aslibekyan[19], Mary F. Feitosa[20], Minjung Kho[21], Solomon K. Musani[22], Melissa Richard[23], Heming Wang[24], Zhe Wang[14], Traci M. Bartz[25], Lawrence F. Bielak[21], Archie Campbell[26], Rajkumar Dorajoo[27], Virginia Fisher[28], Fernando P. Hartwig[29,30], Andrea R.V.R. Horimoto[31], Changwei Li[32], Kurt K. Lohman[33], Jonathan Marten[34], Xueling Sim[35], Albert V. Smith[36,37], Salman M. Tajuddin[38], Maris Alver[39], Marzyeh Amini[40], Mathilde Boissel[41], Jin Fang Chai[35], Xu Chen[42], Jasmin Divers[43], Evangelos Evangelou[44,45], Chuan Gao[46], Mariaelisa Graff[16], Sarah E. Harris[26,47], Meian He[48], Fang-Chi Hsu[43], Anne U. Jackson[49], Jing Hua Zhao[50], Aldi T. Kraja[20], Brigitte Kühnel[51,52], Federica Laguzzi[53], Leo-Pekka Lyytikäinen[54,55], Ilja M. Nolte[40], Rainer Rauramaa[56], Muhammad Riaz[57], Antonietta Robino[58], Rico Rueedi[59,60], Heather M. Stringham[49], Fumihiko Takeuchi[61], Peter J. van der Most[40], Tibor V. Varga[62], Niek Verweij[63], Erin B. Ware[64], Wanqing Wen[65], Xiaoyin Li[66], Lisa R. Yanek[67], Najaf Amin[18], Donna K. Arnett[68], Eric Boerwinkle[14,69], Marco Brumat[70], Brian Cade[24], Mickaël Canouil[41], Yii-Der Ida Chen[17], Maria Pina Concas[58], John Connell[71], Renée de Mutsert[72], H. Janaka de Silva[73], Paul S. de Vries[14], Ayşe Demirkan[18], Jingzhong Ding[74], Charles B. Eaton[75], Jessica D. Faul[64], Yechiel Friedlander[76], Kelley P. Gabriel[77], Mohsen Ghanbari[18,78], Franco Giulianini[7], Chi Charles Gu[5], Dongfeng Gu[79], Tamara B. Harris[80], Jiang He[81,82], Sami Heikkinen[83,84], Chew-Kiat Heng[85,86], Steven C. Hunt[87,88], M. Arfan Ikram[18,89], Jost B. Jonas[90,91], Woon-Puay Koh[35,92], Pirjo Komulainen[56], Jose E. Krieger[31], Stephen B. Kritchevsky[74], Zoltán Kutalik[60,93], Johanna Kuusisto[84], Carl D. Langefeld[43], Claudia Langenberg[50], Lenore J. Launer[80], Karin Leander[53], Rozenn N. Lemaitre[94], Cora E. Lewis[95], Jingjing Liang[66], Lifelines Cohort Study, Jianjun Liu[27,96], Reedik Mägi[39], Ani Manichaikul[97], Thomas Meitinger[98,99], Andres Metspalu[39], Yuri Milaneschi[100], Karen L. Mohlke[101], Thomas H. Mosley Jr.[102], Alison D. Murray[103], Mike A. Nalls[104,105], Ei-Ei Khaing Nang[35], Christopher P. Nelson[106,107], Sotoodehnia Nona[108], Jill M. Norris[109], Chiamaka Vivian Nwuba[1], Jeff O'Connell[110,111], Nicholette D. Palmer[112], George J. Papanicolau[113], Raha Pazoki[44], Nancy L. Pedersen[42], Annette Peters[52,114], Patricia A. Peyser[21], Ozren Polasek[115,116,117], David J. Porteous[26,47], Alaitz Poveda[62], Olli T. Raitakari[118,119], Stephen S. Rich[97], Neil Risch[120], Jennifer G. Robinson[121], Lynda M. Rose[7], Igor Rudan[122], Pamela J. Schreiner[123], Robert A. Scott[50], Stephen S. Sidney[124], Mario Sims[22], Jennifer A. Smith[21,64], Harold Snieder[40], Tamar Sofer[10,24], John M. Starr[47,125], Barbara Sternfeld[124], Konstantin Strauch[126,127], Hua Tang[128], Kent D. Taylor[17], Michael Y. Tsai[129], Jaakko Tuomilehto[130,131], André G. Uitterlinden[132], M. Yldau van der Ende[63], Diana van Heemst[4], Trudy Voortman[18], Melanie Waldenberger[51,52], Patrik Wennberg[133], Gregory Wilson[134], Yong-Bing Xiang[135], Jie Yao[17], Caizheng Yu[48], Jian-Min Yuan[136,137], Wei Zhao[21], Alan B. Zonderman[138], Diane M. Becker[67], Michael Boehnke[49], Donald W. Bowden[112], Ulf de Faire[53], Ian J. Deary[47,139], Paul Elliott[44,140], Tõnu Esko[39,141], Barry I. Freedman[142], Philippe Froguel[41,143], Paolo Gasparini[58,70], Christian Gieger[51,144], Norihiro Kato[61], Markku Laakso[84], Timo A. Lakka[56,83,145], Terho Lehtimäki[54,55], Patrik K.E. Magnusson[42], Albertine J. Oldehinkel[146], Brenda W.J.H. Penninx[100], Nilesh J. Samani[106,107], Xiao-Ou Shu[65], Pim van der Harst[63,147,148], Jana V. Van Vliet-Ostaptchouk[149], Peter Vollenweider[150], Lynne E. Wagenknecht[151], Ya X. Wang[91], Nicholas J. Wareham[50], David R. Weir[64], Tangchun Wu[48], Wei Zheng[65], Xiaofeng Zhu[66], Michele K. Evans[38], Paul W. Franks[62,133,152,153], Vilmundur Gudnason[36,154], Caroline Hayward[34], Bernardo L. Horta[29], Tanika N. Kelly[81], Yongmei Liu[155], Kari E. North[16], Alexandre C. Pereira[31], Paul M. Ridker[7,8], E. Shyong Tai[35,92,156], Rob M. van Dam[35,156],

Ervin R. Fox[157], Sharon L.R. Kardia[21], Ching-Ti Liu [28], Dennis O. Mook-Kanamori[72,158], Michael A. Province[20], Susan Redline[24], Cornelia M. van Duijn[18], Jerome I. Rotter[17], Charles B. Kooperberg[159], W. James Gauderman[160], Bruce M. Psaty[124,161], Kenneth Rice [162], Patricia B. Munroe [11,163], Myriam Fornage[23], L. Adrienne Cupples[28,164], Charles N. Rotimi[3], Alanna C. Morrison[14], Dabeeru C. Rao[5] & Ruth J.F. Loos [165,166]

[1]Novo Nordisk Foundation Center for Basic Metabolic Research, Faculty of Health and Medical Sciences, University of Copenhagen, Copenhagen 2200, Denmark. [2]Department of Environmental Medicine and Public Health, The Icahn School of Medicine at Mount Sinai, New York 10029 NY, USA. [3]Center for Research on Genomics and Global Health, National Human Genome Research Institute, National Institutes of Health, Bethesda 20892 MD, USA. [4]Internal Medicine, Gerontology and Geriatrics, Leiden University Medical Center, Leiden 2300 RC, The Netherlands. [5]Division of Biostatistics, Washington University School of Medicine, St. Louis 63110 MO, USA. [6]Department of Genetic Epidemiology, University of Regensburg, Regensburg 93051, Germany. [7]Preventive Medicine, Brigham and Women's Hospital, Boston 02215 MA, USA. [8]Harvard Medical School, Boston 02131 MA, USA. [9]Clinical and Translational Epidemiology Unit, Massachusetts General Hospital, Boston 02114 MA, USA. [10]Department of Medicine, Harvard Medical School, Boston 02115 MA, USA. [11]Clinical Pharmacology, William Harvey Research Institite, Barts and The London School of Medicine and Dentistry, Queen Mary University of London, London EC1M 6BQ, UK. [12]Department of Epidemiology, Harvard School of Public Health, Boston 02115 MA, USA. [13]Centre de Bioinformatique, Biostatistique et Biologie Intégrative (C3BI), Institut Pasteur, Paris 75015, France. [14]Human Genetics Center, Department of Epidemiology, Human Genetics, and Environmental Sciences, School of Public Health, The University of Texas Health Science Center at Houston, Houston 77030 TX, USA. [15]Cardiovascular Division, Department of Medicine, Washington University, St. Louis 63110 MO, USA. [16]Epidemiology, University of North Carolina Gillings School of Global Public Health, Chapel Hill 27514 NC, USA. [17]The Institute for Translational Genomics and Population Sciences, Division of Genomic Outcomes, Department of Pediatrics, Los Angeles Biomedical Research Institute at Harbor-UCLA Medical Center, Torrance 90502 CA, USA. [18]Department of Epidemiology, Erasmus University Medical Center, Rotterdam 3015 CE, The Netherlands. [19]Department of Epidemiology, University of Alabama at Birmingham, Birmingham 35294 AL, USA. [20]Division of Statistical Genomics, Department of Genetics, Washington University School of Medicine, St. Louis 63108 MO, USA. [21]Department of Epidemiology, School of Public Health, University of Michigan, Ann Arbor 48109 MI, USA. [22]Jackson Heart Study, Department of Medicine, University of Mississippi Medical Center, Jackson 39213 MS, USA. [23]Institute of Molecular Medicine, McGovern Medical School, University of Texas Health Science Center at Houston, Houston 77030 TX, USA. [24]Division of Sleep and Circadian Disorders, Brigham and Women's Hospital, Boston 02115 MA, USA. [25]Cardiovascular Health Research Unit, Biostatistics and Medicine, University of Washington, Seattle 98101 WA, USA. [26]Centre for Genomic & Experimental Medicine, Institute of Genetics & Molecular Medicine, University of Edinburgh, Edinburgh EH4 2XU, UK. [27]Genome Institute of Singapore, Agency for Science Technology and Research, Singapore 138672, Singapore. [28]Biostatistics, Boston University School of Public Health, Boston 02118 MA, USA. [29]Postgraduate Program in Epidemiology, Federal University of Pelotas, Pelotas 96020220 RS, Brazil. [30]Medical Research Council Integrative Epidemiology Unit, University of Bristol, Bristol BS8 2BN, UK. [31]Laboratory of Genetics and Molecular Cardiology, Heart Institute (InCor), University of São Paulo Medical School, São Paulo 01246903 SP, Brazil. [32]Epidemiology and Biostatistics, University of Giorgia at Athens College of Public Health, Athens 30602 GA, USA. [33]Public Health Sciences, Biostatistical Sciences, Wake Forest University Health Sciences, Winston-Salem 27157 NC, USA. [34]Medical Research Council Human Genetics Unit, Institute of Genetics and Molecular Medicine, Institute of Genetics and Molecular Medicine, University of Edinburgh, Edinburgh EH4 2XU, UK. [35]Saw Swee Hock School of Public Health, National University Health System and National University of Singapore, Singapore 117549, Singapore. [36]Icelandic Heart Association, 201 Kopavogur, Iceland. [37]Department of Biostatistics, University of Michigan, Ann Arbor 48109 MI, USA. [38]Health Disparities Research Section, Laboratory of Epidemiology and Population Sciences, National Institute on Aging, National Institutes of Health, Baltimore 21224 MD, USA. [39]Estonian Genome Center, University of Tartu, Tartu 51010, Estonia. [40]Department of Epidemiology, University of Groningen, University Medical Center Groningen, Groningen 9700 RB, The Netherlands. [41]CNRS UMR 8199, European Genomic Institute for Diabetes (EGID), Institut Pasteur de Lille, University of Lille, Lille 59000, France. [42]Department of Medical Epidemiology and Biostatistics, Karolinska Institutet, Stockholm, Stockholm 17177, Sweden. [43]Department of Biostatistical Sciences, Wake Forest School of Medicine, Winston-Salem 27157 NC, USA. [44]Department of Epidemiology and Biostatistics, Imperial College London, London W2 1PG, UK. [45]Department of Hygiene and Epidemiology, University of Ioannina Medical School, Ioannina 45110, Greece. [46]Molecular Genetics and Genomics Program, Wake Forest School of Medicine, Winston-Salem 27157 NC, USA. [47]Centre for Cognitive Ageing and Cognitive Epidemiology, The University of Edinburgh, Edinburgh EH8 9JZ, UK. [48]Department of Occupational and Environmental Health and State Key Laboratory of Environmental Health for Incubating, Tongji Medical College, Huazhong University of Science and Technology, Wuhan 430014, China. [49]Department of Biostatistics and Center for Statistical Genetics, University of Michigan, Ann Arbor 48109 MI, USA. [50]MRC Epidemiology Unit, University of Cambridge, Cambridge CB2 0QQ, UK. [51]Research Unit of Molecular Epidemiology, Helmholtz Zentrum München, German Research Center for Environmental Health, Neuherberg 85764, Germany. [52]Institute of Epidemiology, Helmholtz Zentrum München, German Research Center for Environmental Health, Neuherberg 85764, Germany. [53]Unit of Cardiovascular Epidemiology, Institute of Environmental Medicine, Karolinska Institutet, Stockholm 17177, Sweden. [54]Department of Clinical Chemistry, Fimlab Laboratories, Tampere 33014, Finland. [55]Department of Clinical Chemistry, Finnish Cardiovascular Research Center—Tampere, Faculty of Medicine and Life Sciences, University of Tampere, Tampere 33014, Finland. [56]Foundation for Research in Health Exercise and Nutrition, Kuopio Research Institute of Exercise Medicine, Kuopio 70100, Finland. [57]College of Medicine, Biological Sciences and Psychology, Health Sciences, The Infant Mortality and Morbidity Studies (TIMMS), Leicester LE1 7RH, UK. [58]Institute for Maternal and Child Health—IRCCS "Burlo Garofolo", Trieste 34137, Italy. [59]Department of Computational Biology, University of Lausanne, Lausanne 1015, Switzerland. [60]Swiss Institute of Bioinformatics, 1015 Lausanne, Switzerland. [61]Department of Gene Diagnostics and Therapeutics, Research Institute, National Center for Global Health and Medicine, Tokyo 1628655, Japan. [62]Department of Clinical Sciences, Genetic and Molecular Epidemiology Unit, Lund University Diabetes Centre, Skåne University Hospital, Malmö 20502, Sweden. [63]University of Groningen, University Medical Center Groningen, Department of Cardiology, Groningen 9700 RB, The Netherlands. [64]Survey Research Center, Institute for Social Research, University of Michigan, Ann Arbor 48104 MI, USA. [65]Division of Epidemiology, Department of Medicine, Vanderbilt University School of Medicine, Nashville 37203 TN, USA. [66]Department of Population and Quantitative Health Sciences, Case Western Reserve University, Cleveland 44106 OH, USA. [67]Division of General Internal Medicine, Department of Medicine, Johns Hopkins University School of Medicine, Baltimore 21287 MD, USA. [68]Dean's Office, University of Kentucky College of Public Health, Lexington 40536 KY, USA. [69]Human Genome Sequencing Center, Baylor College of Medicine, Houston 77030 TX, USA. [70]Department of Medical Sciences, University of Trieste, Trieste 34137, Italy. [71]Ninewells Hospital & Medical School, University of Dundee, Dundee DD1 9SY Scotland, UK. [72]Clinical Epidemiology, Leiden University Medical Center, Leiden 2300 RC, Netherlands. [73]Department of Medicine, Faculty of Medicine, University of Kelaniya, Ragama 11600, Sri Lanka. [74]Department of Internal Medicine, Section on

Gerontology and Geriatric Medicine, Wake Forest School of Medicine, Winston-Salem 27157 NC, USA. [75]Department of Family Medicine and Epidemiology, Alpert Medical School of Brown University, Providence 02860 RI, USA. [76]Braun School of Public Health, Hebrew University-Hadassah Medical Center, Jerusalem 91120, Israel. [77]Department of Epidemiology, Human Genetics & Environmental Sciences, School of Public Health, The University of Texas Health Science Center at Austin, Austin 78712 TX, USA. [78]Department of Genetics, School of Medicine, Mashhad University of Medical Sciences, Mashhad 91778-99191, Iran. [79]Department of Epidemiology, State Key Laboratory of Cardiovascular Disease, Fuwai Hospital, National Center for Cardiovascular Diseases, Chinese Academy of Medical Sciences and Peking Union Medical College, Beijing 100006, China. [80]Laboratory of Epidemiology and Population Sciences, National Institute on Aging, National Institutes of Health, Bethesda 20892 MD, USA. [81]Epidemiology, Tulane University School of Public Health and Tropical Medicine, New Orleans 70112 LA, USA. [82]Medicine, Tulane University School of Medicine, New Orleans 70112 LA, USA. [83]Institute of Biomedicine, School of Medicine, University of Eastern Finland, Kuopio Campus 70211, Finland. [84]Institute of Clinical Medicine, Internal Medicine, University of Eastern Finland, Kuopio 70210, Finland. [85]Department of Paediatrics, Yong Loo Lin School of Medicine, National University of Singapore, Singapore 119228, Singapore. [86]Khoo Teck Puat—National University Children's Medical Institute, National University Health System, Singapore 119228, Singapore. [87]Division of Epidemiology, Department of Internal Medicine, University of Utah, Salt Lake City 84132 UT, USA. [88]Department of Genetic Medicine, Weill Cornell Medicine, Doha 24144, Qatar. [89]Department of Radiology and Nuclear Medicine, Erasmus University Medical Center, Rotterdam 3015 GD, The Netherlands. [90]Department of Ophthalmology, Medical Faculty Mannheim, University Heidelberg, Mannheim 68167, Germany. [91]Beijing Institute of Ophthalmology, Beijing Tongren Eye Center, Beijing Ophthalmology and Visual Science Key Lab, Beijing Tongren Hospital, Capital Medical University, Beijing 100730, China. [92]Health Services and Systems Research, Duke-NUS Medical School, Singapore 169857, Singapore. [93]Institute of Social and Preventive Medicine, Lausanne University Hospital, Lausanne 1010, Switzerland. [94]Cardiovascular Health Research Unit, Medicine, University of Washington, Seattle 98101 WA, USA. [95]Division of Preventive Medicine, Department of Medicine, University of Alabama at Birmingham, School of Medicine, Birmingham 35294 AL, USA. [96]Department of Ophthalmology, Yong Loo Lin School of Medicine, National University of Singapore, Singapore 117597, Singapore. [97]Center for Public Health Genomics, University of Virginia School of Medicine, Charlottesville 22908 VA, USA. [98]Institute of Human Genetics, Helmholtz Zentrum München, German Research Center for Environmental Health, Neuherberg 85764, Germany. [99]Institute of Human Genetics, Technische Universität München, Munich 80333, Germany. [100]Department of Psychiatry, Amsterdam Neuroscience and Amsterdam Public Health Research Institute, VU University Medical Center, Amsterdam 1081 HV, The Netherlands. [101]Department of Genetics, University of North Carolina, Chapel Hill 27514 NC, USA. [102]Geriatrics, Medicine, University of Mississippi, Jackson 39216 MS, USA. [103]The Institute of Medical Sciences, Aberdeen Biomedical Imaging Centre, University of Aberdeen, Aberdeen AB25 2ZD, UK. [104]Molecular Genetics Section, Laboratory of Neurogenetics, National Institute on Aging, Bethesda 20892 MD, USA. [105]Data Tecnica International, Glen Echo 20812 MD, USA. [106]Department of Cardiovascular Sciences, University of Leicester, Leicester LE3 9PQ, UK. [107]NIHR Leicester Biomedical Research Centre, Glenfield Hospital, Leicester LE3 9QD, UK. [108]Cardiovascular Health Research Unit, Division of Cardiology, University of Washington, Seattle 98101 WA, USA. [109]Department of Epidemiology, University of Colorado Denver, Aurora 80045 CO, USA. [110]Division of Endocrinology, Diabetes, and Nutrition, University of Maryland School of Medicine, Baltimore 21201 MD, USA. [111]Program for Personalized and Genomic Medicine, University of Maryland School of Medicine, Baltimore 21201 MD, USA. [112]Department of Biochemistry, Wake Forest School of Medicine, Winston-Salem 27157 NC, USA. [113]Epidemiology Branch, National Heart, Lung, and Blood Institute, National Institutes of Health, Bethesda 20892 MD, USA. [114]DZHK (German Centre for Cardiovascular Research), partner site Munich Heart Alliance, Neuherberg 85764, Germany. [115]Department of Public Health, Department of Medicine, University of Split, Split 21000, Croatia. [116]Psychiatric Hospital "Sveti Ivan", Zagreb 10000, Croatia. [117]Gen-Info Ltd., 10000 Zagreb, Croatia. [118]Department of Clinical Physiology and Nuclear Medicine, Turku University Hospital, Turku 20521, Finland. [119]Research Centre of Applied and Preventive Cardiovascular Medicine, University of Turku, Turku 20520, Finland. [120]Institute for Human Genetics, Department of Epidemiology and Biostatistics, University of California, San Francisco 94143 CA, USA. [121]Department of Epidemiology and Medicine, University of Iowa, Iowa City 52242 IA, USA. [122]Centre for Global Health Research, Usher Institute of Population Health Sciences and Informatics, University of Edinburgh, Edinburgh EH16 4UX, UK. [123]Division of Epidemiology & Community Health, School of Public Health, University of Minnesota, Minneapolis 55454 MN, USA. [124]Kaiser Permanente Washington, Health Research Institute, Seattle 98101 WA, USA. [125]Alzheimer Scotland Dementia Research Centre, The University of Edinburgh, Edinburgh EH8 9JZ, UK. [126]Institute of Genetic Epidemiology, Helmholtz Zentrum München, German Research Center for Environmental Health, Neuherberg 85764, Germany. [127]Institute of Medical Informatics Biometry and Epidemiology, Ludwig-Maximilians-Universitat Munchen, Munich 81377, Germany. [128]Department of Genetics, Stanford University, Stanford 94305 CA, USA. [129]Department of Laboratory Medicine and Pathology, University of Minnesota, Minneapolis 55455 MN, USA. [130]Public Health Solutions, National Institute for Health and Welfare, Helsinki 00271, Finland. [131]Diabetes Research Group, King Abdulaziz University, Jeddah 21589, Saudi Arabia. [132]Department of Internal Medicine, Erasmus University Medical Center, Rotterdam 3015 CE, The Netherlands. [133]Department of Public Health & Clinical Medicine, Umeå University, Umeå 90185 Västerbotten, Sweden. [134]Jackson Heart Study, School of Public Health, Jackson State University, Jackson 39213 MS, USA. [135]State Key Laboratory of Oncogene and Related Genes & Department of Epidemiology, Shanghai Cancer Institute, Renji Hospital, Shanghai Jiaotong University School of Medicine, Shanghai 200000, China. [136]Department of Epidemiology, Graduate School of Public Health, University of Pittsburgh, Pittsburgh 15261 PA, USA. [137]Division of Cancer Control and Population Sciences, UPMC Hillman Cancer, University of Pittsburgh, Pittsburgh 15232 PA, USA. [138]Behavioral Epidemiology Section, Laboratory of Epidemiology and Population Sciences, National Institute on Aging, National Institutes of Health, Baltimore 21224 MD, USA. [139]Psychology, The University of Edinburgh, Edinburgh EH8 9JZ, UK. [140]MRC-PHE Centre for Environment and Health, Imperial College London, London W2 1PG, UK. [141]Broad Institute of the Massachusetts Institute of Technology and Harvard University, Boston 02142 MA, USA. [142]Section on Nephrology, Department of Internal Medicine, Wake Forest School of Medicine, Winston-Salem 27157 NC, USA. [143]Department of Genomics of Common Disease, Imperial College London, London W12 0NN, UK. [144]German Center for Diabetes Research (DZD e.V.), Neuherberg 85764, Germany. [145]Department of Clinical Physiology and Nuclear Medicine, Kuopio University Hospital, Kuopio 70210, Finland. [146]Department of Psychiatry, University of Groningen, University Medical Center Groningen, Groningen 9713 GZ, The Netherlands. [147]Department of Genetics, University of Groningen, University Medical Center Groningen, Groningen 9700 RB, The Netherlands. [148]Durrer Center for Cardiogenetic Research, ICIN-Netherlands Heart Institute, Utrecht 1105 AZ, The Netherlands. [149]Department of Endocrinology, University of Groningen, University Medical Center Groningen, Groningen 9713 GZ, The Netherlands. [150]Internal Medicine, Department of Medicine, Lausanne University Hospital, Lausanne 1011, Switzerland. [151]Public Health Sciences, Wake Forest School of Medicine, Winston-Salem 27157 NC, USA. [152]Harvard T. H. Chan School of Public Health, Department of Nutrition, Harvard University, Boston 02115 MA, USA. [153]OCDEM, Radcliffe Department of Medicine, University of Oxford, Oxford OX3 7LE, UK. [154]Faculty of Medicine, University of Iceland, Reykjavik 101, Iceland. [155]Public Health Sciences, Epidemiology and Prevention, Wake Forest University Health Sciences, Winston-Salem 27157 NC, USA. [156]Department of Medicine, Yong Loo Lin School of Medicine, National University of Singapore, Singapore 119228, Singapore. [157]Cardiology, Medicine, University of Mississippi Medical Center, Jackson 39216 MS, USA. [158]Public Health and Primary Care, Leiden University Medical Center, Leiden 2300 RC, The Netherlands. [159]Fred Hutchinson Cancer Research Center, University of Washington School of Public Health, Seattle 98109 WA, USA. [160]Biostatistics, Preventive Medicine, University of Southern California, Los Angeles 90032 CA, USA. [161]Cardiovascular Health Research Unit,

Epidemiology, Medicine and Health Services, University of Washington, Seattle 98101 WA, USA. [162]Department of Biostatistics, University of Washington, Seattle 98105 WA, USA. [163]NIHR Barts Cardiovascular Research Centre, Barts and The London School of Medicine and Dentistry, Queen Mary University of London, Charterhouse Square, London EC1M 6BQ, UK. [164]NHLBI Framingham Heart Study, Framingham 01702 MA, USA. [165]Icahn School of Medicine at Mount Sinai, The Charles Bronfman Institute for Personalized Medicine, New York 10029 NY, USA. [166]Icahn School of Medicine at Mount Sinai, The Mindich Child Health and Development Institute, New York 10029 NY, USA

## Lifelines Cohort Study

Behrooz Z. Alizadeh[40], H. Marike Boezen[40], Lude Franke[147], Gerjan Navis[167], Marianne Rots[168], Morris Swertz[147], Bruce H.R. Wolffenbuttel[149] & Cisca Wijmenga[147]

[167]Department of Internal Medicine, Division of Nephrology, University of Groningen, University Medical Center Groningen, Groningen 9713 GZ, The Netherlands. [168]Department of Medical Biology, University of Groningen, University Medical Center Groningen, Groningen 9713 GZ, The Netherlands

