## [Peer Review File · Nature Communications]

Reviewer #1 (Remarks to the Author):

The authors presented an interesting study which investigated the potential interaction between physical activity and genetic variants on lipids levels. The question is very important, as lipids is a major modifiable risk factors for coronary artery diseases. It is important to investigate whether life style changes could potentially interact with the genetic variants to reduce cholesterol levels.

This is a carefully conducted study. The statistics used are well justified. I only have one minor comment for the authors to address:

when studying lipids, it is always important to connect them to their impact on CAD. The study should not be different. Recent studies [Khera NEJM 2016] showed that healthy life style interacts with CAD genetic risk in a very significant way, where individuals with good life style can substantially reduce the CAD risk even if they have high genetic risk. I wonder if the authors could study the interaction between the PA and the genetic risk score for CAD and for each lipid levels. Especially, it is of interest to understand if the PA can be sufficient to modify the genetic risk for high cholesterol levels.

Reviewer #2 (Remarks to the Author):

T. Kilpelainen and colleagues investigated SNPs associated with lipid levels in interaction with physical activity in up to 250K participants of diverse ancestries. They identified four novel loci missed by conventional GWAS approaches. This is an original and timely study.

1- Total cholesterol is routinely analyzed in GWAS for lipid traits. The authors may also analyze this trait or at least justify why they focus on HDL-cholesterol, LDL-cholesterol and triglycerides in this study.

2-The definition used to classify participants as active (> 1 h or moderate physical activity) or inactive (\leq 1h of moderate physical activity) is quite arbitrary. As a result people with moderate and high levels of physical activity are considered as physically active. The authors may consider using 3 physical activity categories (low, moderate, high), as routinely done in literature (Reddon et al., Sci Rep 2016).

3-The authors investigated interactions between SNPs and physical activity on lipid levels. As physical activity and sedentary behaviors are independent predictors of cardio-metabolic outcomes, they may study interactions between SNPs and sedentary behaviors (e.g. number of hours of TV watching) on lipids as well.

4-Did the authors consider using UKBiobank to increase the statistical power of the analysis and identify more loci?

5-The authors indicate that 250 loci have been previously associated with lipid levels in literature, but they cite an exome-wide association study. I recommend using a double-blind systematic review of GWAS catalogue and literature databases, followed by a verification of the linkage disequilibrium between reported loci, to establish an up to date list of GWAS-significant lipid loci (Christie, Sci Rep 2017).

6-Surprisingly, the authors did not identify interaction between well-established lipid-associated SNPs and physical activity. This may be due to the modest effect of each SNP on lipids when analyzed separately. The authors may complement this analysis by an interaction test between a cumulative lipid SNP gene score and physical activity, in order to increase the statistical power of the analysis.

7-The authors mention suggestive associations ($P < 5 \times 10^{-6}$) between SNPs and weight loss after

bariatric surgery, or drug-induced liver injury in the discussion. Please remove the two sentences, as these associations do not pass the GWAS Bonferroni threshold ($P < 5 \times 10^{-8}$) and are likely to be random results.

8-Did the authors consider performing eQTL studies in appropriate tissues to confirm the candidacy of certain genes. Publically available data can be used for these analyses.

9-The authors mentioned that the 2df joint tests led to the identification of only one novel locus while 'rediscovering' 100 well-established lipid loci. They also indicated that no significant interaction between SNPs and physical activity on lipids was found for 250 well-established lipid loci. Does it mean that the 2df joint test method does not identify interacting SNPs? Further discussion is needed on the relevance of this method in the context of gene x environment interaction studies.

REVIEWER #1

The authors presented an interesting study which investigated the potential interaction between physical activity and genetic variants on lipids levels. The question is very important, as lipids is a major modifiable risk factors for coronary artery diseases. It is important to investigate whether life style changes could potentially interact with the genetic variants to reduce cholesterol levels.

This is a carefully conducted study. The statistics used are well justified. I only have one minor comment for the authors to address:

when studying lipids, it is always important to connect them to their impact on CAD. The study should not be different. Recent studies [Khera NEJM 2016] showed that healthy life style interacts with CAD genetic risk in a very significant way, where individuals with good life style can substantially reduce the CAD risk even if they have high genetic risk. I wonder if the authors could study the interaction between the PA and the genetic risk score for CAD and for each lipid levels. Especially, it is of interest to understand if the PA can be sufficient to modify the genetic risk for high cholesterol levels.

RESPONSE: We thank the reviewer for the valuable suggestion. The study by Khera et al. [PMID 27959714] showed that genetic and lifestyle factors are independently associated with susceptibility to CAD. Thus, as the reviewer pointed out, a favorable lifestyle can substantially reduce CAD risk even in individuals at high genetic risk. To examine whether the same is true for PA and lipids, we have now examined the interactions of PA with genetic risk scores for CAD, HDL-C, LDL-C, and TG on circulating lipid levels.

To estimate the PA-interaction effect for a multi-SNP score consisting of 62 known European-ancestry SNPs associated with CAD, 236 SNPs associated with HDL-C, 194 SNPs associated with LDL-C, and 161 SNPs associated with TG, on circulating lipid levels, we calculated their combined effect as the weighted sum of the individual SNP coefficients using our GWAS summary results. This method has been described previously in detail by Dastani et al. [PMID 22479202].

Consistent with the findings by Khera et al., we did not find a significant interaction between PA and the genetic risk score for CAD on any of the three lipid traits ($P_{LDL-C}=0.18$; $P_{HDL-C}=0.41$, $P_{TG}=0.54$), or between PA and the genetic risk scores for HDL-C ($P=0.14$), LDL-C ($P=0.77$), or TG ($P=0.86$) on the corresponding lipid traits. Our results suggest that PA can improve lipid levels even in individuals at elevated genetic risk. We report on these new findings on **lines 346-351** of the manuscript and describe the method on **lines 476-484**.

REVIEWER #2

T. Kilpelainen and colleagues investigated SNPs associated with lipid levels in interaction with physical

activity in up to 250K participants of diverse ancestries. They identified four novel loci missed by conventional GWAS approaches. This is an original and timely study.

QUESTION 1: Total cholesterol is routinely analyzed in GWAS for lipid traits. The authors may also analyze this trait or at least justify why they focus on HDL-cholesterol, LDL-cholesterol and triglycerides in this study.

RESPONSE: Total cholesterol has commonly been examined in GWAS for lipid traits. We decided not to include total cholesterol in the present analyses as we aimed to identify loci interacting with physical activity, and physical activity is known to affect HDL cholesterol and LDL cholesterol levels in opposing directions. Thus, the magnitude of interaction effects found for total cholesterol levels would be expected to be intermediate between those found for HDL cholesterol or LDL cholesterol alone. For this reason, to avoid unnecessary tests with compromised power and to spare the analysts from each cohort from excess burden of analysis models to run, we decided to exclude total cholesterol from the final analysis plan.

QUESTION 2: The definition used to classify participants as active (> 1 h or moderate physical activity) or inactive (\leq 1h of moderate physical activity) is quite arbitrary. As a result people with moderate and high levels of physical activity are considered as physically active. The authors may consider using 3 physical activity categories (low, moderate, high), as routinely done in literature (Reddon et al., Sci Rep 2016).

RESPONSE: We agree with the reviewer that using a larger number of categories for physical activity could improve the estimation of the dose-response relationship in the interaction effect between the SNPs and physical activity. However, harmonization of environmental exposures in meta-analyses of gene-environment interactions is challenging, and in particular for physical activity that is assessed and quantified in different ways in epidemiological studies (Line 425, S. Table 7). Nearly all physical activity questionnaires allow distinguishing between individuals who do not engage in any regular physical activity, and those who do, but the wide heterogeneity in questionnaire response categories at higher physical activity levels makes it unattainable to generate consistent and valid sub-categories within the 'active' group. Aiming to amass as large a sample size as possible, we decided to define a simple dichotomous physical activity trait that could be derived in a relatively consistent way in all participating studies, and that would also be as consistent as possible with previous findings on gene-physical activity interactions and the relationship between activity levels and health outcomes (Lines 425-428). Thus, the group 'active' was defined to include all individuals who were regularly active during leisure-time or commuting (>1 h/wk of moderate-intensity physical activity) whereas all other individuals were defined 'inactive'. Previous studies in large-scale individual cohorts have demonstrated that the interaction between physical activity and the *FTO* obesity risk locus, or a BMI-increasing genetic risk score, is most pronounced approximately at this activity level [PMID 20824172, 17942823, 19553294]. The choice of the physical activity threshold is also in line with the more general observation that the benefits of increased physical activity on health outcomes do not increase linearly; the most substantial benefits are attained when moving from complete sedentariness to low levels of activity, whereas additional benefits diminish at higher activity levels [e.g PMID 25733647]. To examine the dose-response curve of gene-

physical activity interactions in more detail, access to very large cohorts with quantitative measures of physical activity will be needed (e.g. the UK Biobank when lipid data become available).

QUESTION 3: The authors investigated interactions between SNPs and physical activity on lipid levels. As physical activity and sedentary behaviors are independent predictors of cardio-metabolic outcomes, they may study interactions between SNPs and sedentary behaviors (e.g. number of hours of TV watching) on lipids as well.

RESPONSE: We agree with the reviewer that it is also important to study interactions between SNPs and sedentary behaviors. We initially considered including an additional exposure variable based on sedentary behavior. However, only about a third of the participating studies had data available on TV viewing time or daily sitting, which was considered too few for having sufficient statistical power to identify SNP-sedentary behavior interactions. Hence, the present analyses focus on physical activity only. We will re-consider the possibility of studying interactions with sedentary behaviors in the next round of meta-analyses where more studies with measures of sedentary behavior may become available (e.g. UK Biobank and Million Veteran Program).

QUESTION 4: Did the authors consider using UKBiobank to increase the statistical power of the analysis and identify more loci?

RESPONSE: We considered including UK Biobank in the present meta-analyses to increase the statistical power and identify more loci. However, while lipid levels are included in the biomarker measurement panel of the UK Biobank, the measurements are still ongoing and thus data on lipid levels are not currently available. For the future, we have plans in place to extend this study of SNPxPA interactions on lipid levels in the next 1-2 years. The extended project will include the UK Biobank and Million Veteran Program, among other large cohorts.

QUESTION 5: The authors indicate that 250 loci have been previously associated with lipid levels in literature, but they cite an exome-wide association study. I recommend using a double-blind systematic review of GWAS catalogue and literature databases, followed by a verification of the linkage disequilibrium between reported loci, to establish an up to date list of GWAS-significant lipid loci (Christie, Sci Rep 2017).

RESPONSE: We thank the reviewer for a valuable suggestion. We have now performed a comprehensive review of known lipid loci and extended a previously curated list of lipid loci by Davis et al. [PMID 29084231], by searching PubMed and Google Scholar databases and screening the GWAS Catalog. After LD pruning by $r^2 < 0.1$ within 1 Mb, the resulting list contained 260, 202, and 185 variants independently associated with HDL-C, LDL-C, and TG, respectively. This procedure and the resulting list of loci have now been described on **lines 476-481** and in **Supplementary Tables 11-13** of the manuscript. We have also revised the number of loci and the citations on **lines 311-312**.

QUESTION 6: Surprisingly, the authors did not identify interaction between well-established lipid-associated SNPs and physical activity. This may be due to the modest effect of each SNP on lipids when analyzed separately. The authors may complement this analysis by an interaction test between a cumulative lipid SNP gene score and physical activity, in order to increase the statistical power of the analysis.

RESPONSE: We agree with the reviewer that in order to improve statistical power, it is also relevant to test the interaction between lipid SNP gene scores and PA. We have now tested the combined PA-interaction of all published European-ancestry lipid associated SNPs on lipid levels, by utilizing our genome-wide summary results for European ancestry. To approximate the combined PA-interaction of all known loci associated with HDL-C, LDL-C, or TG, we calculated their aggregate effect as the weighted sum of the individual SNP coefficients in our GWAS summary results. The approach has been described previously in detail by Dastani et al. [PMID 22479202].

Consistent with our findings for the single SNPs, we did not find a significant interaction between PA and the genetic risk scores for HDL-C ($P=0.14$), LDL-C ($P=0.77$), or TG ($P=0.86$) on the levels of the corresponding lipid traits. We report on these new findings on **lines 346-351** of the manuscript and describe the method on **lines 481-484**.

QUESTION 7: The authors mention suggestive associations ($P < 5 \times 10^{-6}$) between SNPs and weight loss after bariatric surgery, or drug-induced liver injury in the discussion. Please remove the two sentences, as these associations do not pass the GWAS Bonferroni threshold ($P < 5 \times 10^{-8}$) and are likely to be random results.

RESPONSE: As recommended by the reviewer, we have now removed the two sentences from the manuscript.

QUESTION 8: Did the authors consider performing eQTL studies in appropriate tissues to confirm the candidacy of certain genes. Publically available data can be used for these analyses.

RESPONSE: We agree with the reviewer that eQTL studies are important to confirm the candidacy of the nearby genes. Using data from the GTEx portal, we examined associations of the identified loci with gene expression. We describe methods for the eQTL look-ups on **lines 490-494** of the manuscript.

We found that the rs2862183 SNP in *CLASP1* showed an association with *CLASP1* expression in Esophagus Muscularis ($P=2.9 \times 10^{-5}$), and with *TFCP2L1* expression in tibial nerve ($P=4.1 \times 10^{-5}$). The peak *cis*-eQTL for *CLASP1* expression in esophagus muscularis is rs13403769 which is strongly correlated with rs2862183 ($r^2=0.87$). (**Lines 358-361**) Thus, these two variants represent the same signal. The lead *cis*-eQTL associated with *TFCP2L1* expression in tibial nerve is rs35023952 which has only low correlation with rs2862183 ($r^2=0.15$). Thus, the association of rs2862183 with *CLASP1* expression in tibial nerve is explained by low correlation with a much stronger signal in the same region.

We found no eQTL associations for the three other loci we identified. This does not exclude the possibility that the loci are associated with the expression of nearby genes, as these loci were identified due to interaction with PA, and no such data for eQTLs are currently available.

QUESTION 9: The authors mentioned that the 2df joint tests led to the identification of only one novel locus while ‘rediscovering’ 100 well-established lipid loci. They also indicated that no significant interaction between SNPs and physical activity on lipids was found for 250 well-established lipid loci. Does it mean that the 2df joint test method does not identify interacting SNPs? Further discussion is needed on the relevance of this method in the context of gene x environment interaction studies.

RESPONSE: We thank the reviewer for a very relevant remark. As demonstrated by Manning et al. [PMID 21181894], the 2df joint test bolsters the power to detect novel loci when both main and an interaction effect are present. The lack of novel loci identified by the 2df test suggests that the loci showing the strongest SNPxPA interaction on lipid levels are not the same loci that show a strong main effect on lipid levels, as we now describe on **lines 391-394** of the manuscript. Our finding that the 2df joint test mainly identifies main effect lipid loci rather than interactions is consistent with the results published for other human complex traits [e.g. PMID 22581228, 26426971].

STATEMENT ABOUT DATA AVAILABILITY

The meta-analysis summary results generated during the current study will be made available at the CHARGE Summary Results site at [<http://www.chargeconsortium.com/main/results>] at the time of publication, as we state on **lines 495-497** of the manuscript.

Reviewer #1 (Remarks to the Author):

The authors have adequately addressed my comments.

Reviewer #2 (Remarks to the Author):

The authors addressed all my comments, thank you.